# Dopamine encoding of novelty facilitates efficient uncertainty-driven exploration

Yuhao Wang[1], Armin Lak[2], Sanjay G. Manohar[3], Rafal Bogacz[1]*

**1** MRC Brain Network Dynamics Unit, University of Oxford, Oxford, United Kingdom, **2** Department of Physiology, Anatomy and Genetics, University of Oxford, Oxford, United Kingdom, **3** Nuffield Department of Clinical Neurosciences, University of Oxford, Oxford, United Kingdom

* rafal.bogacz@bndu.ox.ac.uk

**Data Availability Statement:** All data analysed in this manuscript and all code used in data analysis and simulations is freely available at: https://github.com/wang-yu-hao/BasalGangliaExploration.

## Abstract

When facing an unfamiliar environment, animals need to explore to gain new knowledge about which actions provide reward, but also put the newly acquired knowledge to use as quickly as possible. Optimal reinforcement learning strategies should therefore assess the uncertainties of these action–reward associations and utilise them to inform decision making. We propose a novel model whereby direct and indirect striatal pathways act together to estimate both the mean and variance of reward distributions, and mesolimbic dopaminergic neurons provide transient novelty signals, facilitating effective uncertainty-driven exploration. We utilised electrophysiological recording data to verify our model of the basal ganglia, and we fitted exploration strategies derived from the neural model to data from behavioural experiments. We also compared the performance of directed exploration strategies inspired by our basal ganglia model with other exploration algorithms including classic variants of upper confidence bound (UCB) strategy in simulation. The exploration strategies inspired by the basal ganglia model can achieve overall superior performance in simulation, and we found qualitatively similar results in fitting model to behavioural data compared with the fitting of more idealised normative models with less implementation level detail. Overall, our results suggest that transient dopamine levels in the basal ganglia that encode novelty could contribute to an uncertainty representation which efficiently drives exploration in reinforcement learning.

## Author summary

Humans and other animals learn from rewards and losses resulting from their actions to maximise their chances of survival. In many cases, a trial-and-error process is necessary to determine the most rewarding action in a certain context. During this process, determining how much resource should be allocated to acquiring information ("exploration") and how much should be allocated to utilising the existing information to maximise reward ("exploitation") is key to the overall effectiveness, i.e., the maximisation of total reward obtained with a certain amount of effort. We propose a theory whereby an area within the mammalian brain called the basal ganglia integrates current knowledge about the mean reward, reward uncertainty and novelty of an action in order to implement an algorithm

**Funding:** This work has been supported by Biotechnology and Biological Sciences Research Council (grant BB/S006338/1 to RB, https://www.ukri.org/councils/bbsrc/); by the Medical Research Council (grant MC_UU_00003/1 to RB, https://www.ukri.org/councils/mrc/); by the Henry Dale Fellowship from the Wellcome Trust (to AL, https://wellcome.org/); by the Royal Society (grant 213465 to AL, https://royalsociety.org/); by the National Institute for Healthcare Research (NIHR) Oxford Biomedical Research Centre (BRC) (to SGM, https://oxfordbrc.nihr.ac.uk/); and by the James S. McDonnell Foundation (to SGM, https://www.jsmf.org/). The funders had no role in study design, data collection and analysis, decision to publish, or preparation of the manuscript.

which optimally allocates resources between exploration and exploitation. We verify our theory using behavioural experiments and electrophysiological recording, and show in simulations that the model also achieves good performance in comparison with established benchmark algorithms.

## Introduction

In order to survive, animals must develop efficient strategies of reinforcement learning to maximise the reward of their actions. An important factor in effective reinforcement learning is optimised modulation of exploration and exploitation. If an animal already possesses knowledge about a safe and nutritious food source, say a fruit, should it prioritise seeking for that familiar fruit in future foraging, or should it keep trying out unfamiliar alternatives?

In this study, we generalise from real-world scenarios and define exploration to be any behaviour by a learning agent that favours actions which are sub-optimal in terms of their expected rewards according to the current best knowledge of the agent, and exploitation as behaviour that chooses the optimal action with highest expected reward. Modulating exploration and exploitation is no trivial task, not least because in real-world scenarios there are often factors such as motivation [1], non-stationarity of the environment [2] and balancing of short-term and long-term reward optimisation [3] that together influence the optimal strategy in complex ways. Here, we focus on a quintessential problem without the additional complexity to establish a feasible neural mechanism for exploration-exploitation modulation based on reward uncertainty estimation.

The problem in question is the classic multi-armed bandit task [4–7]. By design of the task, rewards are simplified to one-dimensional numerical values and actions to having no difference in effort exertion—these simplifications also effectively eliminate the necessity to consider the contribution of sensorimotor error to uncertainty, so that uncertainty is solely a result of the stochasticity in the environment and can be better controlled in experimental design. During the task, the agent has to choose one out of multiple slot machines ("arms of the bandit") to play on each trial. Each of the arms produces a reward represented as a scalar numerical value when played. The rewards from each arm are sampled from a probability distribution associated with the arm, which remains stationary throughout each block of trials. The agent is made aware of the start and end of each block of trials as well as the length of each block, and is instructed to maximise the total reward received within each block. We formalise the mathematical description of the task later in Introduction.

In the context of this task, if an agent follows a greedy strategy [8] that does not involve any exploration at all and always prefers the optimal action according to current knowledge, they would simply play each arm exactly once at the beginning of each block of trials and proceed to always choose the one that returns the highest reward for the rest of the block. The performance of this simple strategy quickly deteriorates as the spreads of the reward distributions get larger. A simple modification of the greedy strategy, often dubbed the $\varepsilon$-greedy strategy [8], adds unmodulated exploration. On each trial, there is a probability of $1 - \varepsilon$ that the agent chooses the empirically optimal action, and a probability of $\varepsilon$ that the agent explores by randomly choosing among the actions with equal probability. We call this unmodulated exploration since the chances of an exploratory choice of action is constant and therefore independent of the level of uncertainty the agent experiences. Such unmodulated exploratory behaviour already improves the robustness of the strategy significantly, but lacks in adaptability.

Finding an optimal strategy for the multi-armed bandit with modulated exploration has been an ongoing quest in the world of statistics since it was first mentioned in the context of sequential analysis [9], and multiple studies [6, 10, 11] discussed optimal strategies that achieve the theoretical asymptotic performance bound [10] under certain constraints. These strategies belong to a class called the upper confidence bound (UCB) algorithm, which computes an uncertainty bonus for each action that modulates exploration. This falls under the category of directed exploration strategies as opposed to random exploration strategies as defined in [12]. A hybrid strategy combining features of directed and random exploration was also proposed and mathematically specified, and these three qualitatively different types of exploration strategies were fitted to human behavioural data from a two-armed bandit experiment [12]. Results show that the hybrid strategy explains human behaviour significantly better. In this work, we take inspiration from the normative modelling of behaviour in [12] and propose a novel model of the basal ganglia which facilitates similar exploration strategies, thus attempt to bridge the gap between algorithmic level study of behaviour and neural implementation.

The novel basal ganglia model is based on a series of studies starting with [13], which proposed that the direct pathway with D1 receptor-expressing neurons and the indirect pathway with D2 receptor-expressing neurons in the striatum can together achieve learning of both expectation and variability of the reward resulting from an action during reinforcement learning. Based on this assumption, tonic dopamine level in the striatum can influence the overall level of risk seeking in behaviour due to the opposite effects dopamine has on D1 and D2 neurons. Specifically, higher dopamine level should result in a stronger preference for more risky actions with more variable outcomes. Experimental evidence consistent with this prediction has been previously reviewed [13]. In this work, we further consider the effect of fast transient changes in dopamine level on decision making. Specifically, we note that the transient activity of dopaminergic neurons can encode novelty [14–19], and show that with the novelty signal provided by dopamine, the basal ganglia circuit modelled can facilitate efficient uncertainty-driven exploration strategies.

Later in Introduction, we introduce the example task used throughout this study and review the normative behavioural models of exploration from [12] in more detail. We also review model of the basal ganglia learning reward uncertainty [13]. In Results, we first show that an extended version of this model can approximate the normative exploration strategies. Next, we compare electrophysiologically recorded activities of dopaminergic neurons in the ventral tegmental area to the form of novelty signal required for efficient exploration according to our model. We then make adjustments to the model to more accurately reflect experimental results, and compare the resulting exploration strategies with the normative strategies [12] when fitted to human behaviour in a bandit task. We also compare the performance of strategies derived from the basal ganglia model with that of other effective strategies in simulated bandit tasks. In Discussion, we compare our model with several other theories [16, 20, 21] on the role of dopaminergic neurons in exploration modulation, and formalise experimental predictions and future directions.

## The multi-armed bandit task

Before introducing reinforcement learning models with uncertainty-driven exploration, we formalise here the nomenclature associated with the multi-armed bandit problem used as the example task throughout this study. On each of the $\tau$ sequential trials (indexed $t \in \{1, 2, \ldots, \tau\}$) within a block, the agent needs to choose one from a total of $K$ available slot machines ("arms" of the bandit, indexed $i \in \{1, 2, \ldots, K\}$) to play. Later in this article, we also use the same subscript $i$ to denote activities of neuronal populations that encode latent variables associated with

the action of choosing a certain arm. The chosen arm on each trial is denoted $c[t] \in \{1, 2, \ldots, K\}$. After the selection is made on each trial, a reward of a certain numerical value is randomly sampled from the reward distribution associated with the selected arm (denoted $R_i$) and presented to the agent.

## Normative strategies of uncertainty-driven exploration

The following strategies for uncertainty-driven exploration (Fig 1) all rely on dynamically updated estimates of mean rewards from each arm, which we denote $Q_i[t]$ for arm $i$ at trial $t$, as well as associated posterior uncertainty levels about the mean estimates, which we denote $\sigma_i[t]$. A conceptually straightforward approach to modelling the updating of these latent variables is with Kalman filtering [12], although the neural implementation of such algorithm is potentially complex [22, 23]. A key contribution of this study is an alternative algorithm for estimating posterior uncertainty with a simpler neural implementation supported by experimental evidence, which we will present in detail in Results. Here, we first introduce the normative exploration strategies based on the learned latent variables of mean estimation and posterior uncertainty.

**Directed exploration: Upper confidence bound (UCB).** With estimations of reward expectation and uncertainty levels for each arm of the bandit learned, the upper confidence bound strategy uses the value utility variable [12]

$$V_{\text{UCB},i}[t] = Q_i[t] + \theta\sigma_i[t] + eZ \tag{1}$$

associated with each arm to make the selection at each trial (Fig 1B). Here $\theta$ and $e$ are weighting parameters and $Z \sim \mathcal{N}(0, 1)$ is a standard Gaussian random variable. The arm with the greatest observed value utility is chosen on each trial. The sum of the first two terms gives the upper bound of a confidence interval for the mean reward estimation (hence "upper confidence bound") and the third term introduces unmodulated stochasticity, which can be

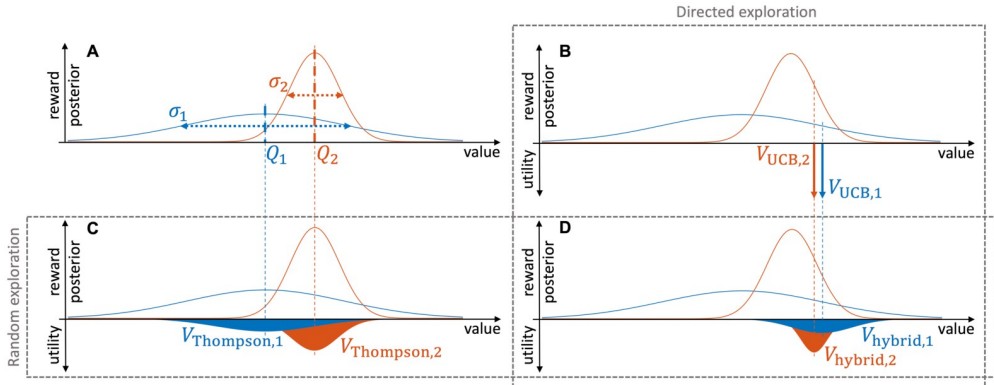

**Fig 1. Demonstration of different types of exploration strategies. The distributions on the upward axis in each panel represent the (Gaussian) posterior estimations of the mean rewards from two arms**. The distributions on the downward axes in panels **B**, **C** and **D** are example distributions of different types of value utility functions (without noise). **A** $Q_1$ and $Q_2$ are posterior means and $\sigma_1$ and $\sigma_2$ are posterior standard deviations, representations of posterior uncertainty levels. **B** With a directed exploration strategy such as UCB, the value utilities (Eq 1) are deterministically biased from the posterior means by an amount proportional to the posterior standard deviation. **C** With a random exploration strategy such as Thompson sampling (in the two-arm case), the value utilities (Eq 4) are sampled around the posterior means with spreads proportional to the posterior standard deviations, so the posterior standard deviations do not bias the action selection, but only modulate the stochasticity. **D** With a hybrid exploration strategy, the value utilities (Eq 27) are sampled around the deterministically biased values of the directed strategy and with spreads proportional to the posterior standard deviations as in the random strategy.

considered as accounting for system noise. Parameter $\theta$ controls the weighting of the "uncertainty bonus", or equivalently the confidence level of the confidence interval. The larger its value, the more optimistic and exploratory the strategy is. In the two-armed case ($K = 2$, $i \in \{1, 2\}$), the probability of choosing arm 1 over 2 is

$$p(c[t] = 1) = p(V_{\mathrm{UCB},1}[t] > V_{\mathrm{UCB},2}[t]) \tag{2}$$

$$= \Phi\left(\frac{Q_1[t] - Q_2[t] + \theta(\sigma_1[t] - \sigma_2[t])}{\sqrt{2e^2}}\right), \tag{3}$$

where $\Phi(\cdot)$ denotes the cumulative density function of the standard Gaussian distribution. This choice probability is dependent on the difference in mean reward estimations and the difference in uncertainty levels ("relative uncertainty" in [12]). Under this strategy, an action currently believed to be less rewarding can actually be the favoured option in terms of choice probability. This is a defining characteristic of a directed exploration strategy, and it generalises to bandit tasks with more than two arms.

**Random exploration: Thompson sampling.**  A different exploration strategy named Thompson sampling [3, 12, 24–26] can be achieved by defining a different value utility (Fig 1C)

$$V_{\mathrm{Thompson},i}[t] = Q_i[t] + \gamma\sigma_i[t]Z. \tag{4}$$

Instead of using the uncertainty level as a deterministic bonus, Thompson sampling samples from a posterior distribution defined by the estimated mean and uncertainty. The specific formalisation here assumes a Gaussian posterior of the form $\mathcal{N}(Q_i[t], \gamma\sigma_i[t])$. Similar to $\theta$ in Eq 1, the parameter $\gamma$ controls the weighting of uncertainty levels by scaling the standard deviation of the Gaussian posterior. In the two-armed ($K = 2$) case, the probability of choosing arm 1 over 2 under Thompson sampling is then

$$p(c[t] = 1) = p(V_{\mathrm{Thompson},1}[t] > V_{\mathrm{Thompson},2}[t]) \tag{5}$$

$$= \Phi\left(\frac{Q_1[t] - Q_2[t]}{\sqrt{\gamma^2(\sigma_1^2[t] + \sigma_2^2[t])}}\right). \tag{6}$$

This probability is again dependent on the difference in mean reward estimations, and also dependent on the sum in uncertainty levels rather than the difference. Thus, the action with higher estimated mean reward is always favoured in terms of choice probability. This is the defining characteristic of random exploration strategies [12]. However, when Thompson sampling is applied to a bandit task with more than two arms, this property does not generalise, and therefore Thompson sampling is not strictly a random exploration strategy in this more general case.

**Hybrid exploration strategy.**  Using regression analysis and model fitting on behavioural data, it has been shown [3, 12] that humans employ an uncertainty-driven strategy that shows characteristics of both directed and random exploration during the bandit task. The choice probability

$$p(c[t] = 1) = \Phi\left(\gamma\frac{Q_1[t] - Q_2[t]}{\sqrt{\sigma_1^2[t] + \sigma_2^2[t]}} + \theta(\sigma_1[t] - \sigma_2[t])\right) \tag{7}$$

was used in [12] to represent such a hybrid strategy. Here, a change in either total uncertainty or relative uncertainty independent of the other can influence action selection through either the sum of uncertainty levels on the denominator or the difference of uncertainty levels on the

numerator, respectively. Note that this choice probability cannot be derived from explicit value utility variables (analogous to those given by Eqs 1 and 4, Fig 1D) associated with each of the arms.

## The basal ganglia model

The normative strategies presented above have so far not been connected to biological implementations. In this study, we show that basal ganglia circuits could potentially support a mechanism for both belief updating and producing value utilities for action selection. We first briefly review a previously described model, and in Results we show the necessary extensions to allow exploration strategies.

**Learning the mean and spread of reward distribution.** It has been previously proposed [13] that, by utilising both direct and indirect striatal pathways that include D1 and D2 receptor-expressing neurons respectively, mean and spread (specifically the mean deviation) of the reward distribution associated with a certain action can be learned simultaneously in the basal ganglia. According to the model, the neural circuit containing both pathways (Fig 2) takes an input representing an available action at the current state from the cortex. The direct pathway with D1 neurons has an excitatory effect on the thalamus, while the indirect pathway with D2 neurons has an inhibitory effect. The combined effects of both pathways in the thalamus represent the current value utility of the action. In a task involving action selection, multiple parallel circuits are required to represent all available actions. The model assumes that belief updating or learning occurs in the weights of corticostriatal projections. We denote the weights of projections from the cortex to the D1 neurons in the direct striatal pathway and D2 neurons in the indirect pathway $G_i[t]$ ($G$ for "GO") and $N_i[t]$ ($N$ for "NO-GO"), respectively, based on the effects of the two pathways on the thalamus. This follows the naming convention of [27], who first proposed the OpAL model which is closely connected to the model we describe in this work. Subscript $i$ indicates the action (choice of arm in the bandit task) being encoded, and $t$ denotes the trial number within a block of trials.

Learning rules in this circuit have been extensively discussed previously [1, 13, 28]. One basic version can be written as

$$\delta[t] = R_{c[t]} - \frac{1}{2}\left(G_{c[t]}[t] - N_{c[t]}[t]\right), \tag{8}$$

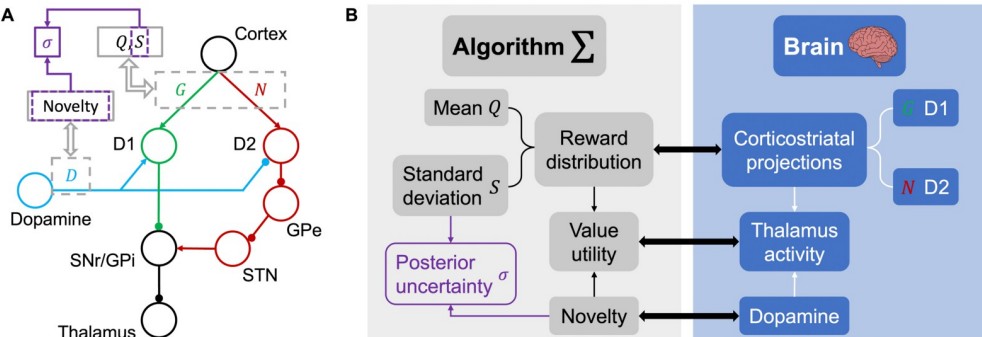

**Fig 2. Illustration of the basal ganglia model. A** Circuit diagram representing the basal ganglia, adapted from [1]. D1/D2 receptor-expressing neurons are involved in direct and indirect striatal pathways, respectively, and dopamine has opposite effects on the two pathways. Both pathways project to the thalamus. **B** Mapping of learned latent parameters in the proposed algorithm (left) onto neural metrics (right).

$$G_{c[t]}[t + 1] = G_{c[t]}[t] + \alpha_{c[t]}[t] f_\epsilon(\delta[t]) - \beta G_{c[t]}[t], \tag{9}$$

$$N_{c[t]}[t + 1] = N_{c[t]}[t] + \alpha_{c[t]}[t] f_\epsilon(-\delta[t]) - \beta N_{c[t]}[t], \tag{10}$$

where $f_\epsilon(x) = x$ for $x > 0$ and $f_\epsilon(x) = \epsilon x$ for $x < 0$ ($0 < \epsilon < 1$). $\delta[t]$ gives the reward prediction error at each trial, as will be shown later. The piecewise linear activation function $f_\epsilon(\cdot)$ over the reward prediction error is an essential element of this learning rule, and evidence shows dopaminergic neurons encoding reward prediction error do exhibit this type of modulation in their responses [29]. The decay terms with scaling parameter $\beta$ keeps the learning variables bounded. Following this learning rule, $G_i$ is a "satisfaction learning" reinforced by better-than-expected outcomes and to a lesser extent diminished by worse-than-expected outcomes, while $N_i$ is a "disappointment learning" reinforced by worse-than-expected outcomes and to a lesser extent diminished by better-than-expected outcomes. $\alpha_i[t]$ is the learning rate parameter taking values in $[0, 1]$ for all values of $i$ and $t$. The substitutions

$$Q_i[t] = (G_i[t] - N_i[t])/2, \tag{11}$$

$$S_i[t] = (G_i[t] + N_i[t])/2 \tag{12}$$

transform the learning rule in Eqs 9 and 10 into

$$Q_{c[t]}[t + 1] = Q_{c[t]}[t] + \alpha_{c[t],q}[t]\delta[t] - \beta Q_{c[t]}[t], \tag{13}$$

$$S_{c[t]}[t + 1] = S_{c[t]}[t] + \alpha_{c[t],s}[t]|\delta[t]| - \beta S_{c[t]}[t], \tag{14}$$

where $\alpha_{i,q}[t] = \alpha_i[t](1 + \epsilon)/2$ and $\alpha_{i,s}[t] = \alpha_i[t](1 - \epsilon)/2$.

Further idealisation of this learning rule gives

$$\delta[t] = R_{c[t]} - Q_{c[t]}[t], \tag{15}$$

$$Q_{c[t]}[t + 1] = Q_{c[t]}[t] + \alpha_{c[t],q}[t]\delta[t], \tag{16}$$

$$S_{c[t]}[t + 1] = S_{c[t]}[t] + \alpha_{c[t],s}[t](|\delta[t]| - S_{c[t]}[t]) \tag{17}$$

as seen in [28], with the constraint $\alpha_{i,q}[t] > \alpha_{i,s}[t]$. Here, we simply set these to be constant values across the experiments for each agent, so that

$$\alpha_{i,q}[t] = \alpha_q, \tag{18}$$

$$\alpha_{i,s}[t] = \alpha_s. \tag{19}$$

Under this learning rule, $Q_i$ and $S_i$ converge to the stationary point

$$Q_i^* = \mathbb{E}\{R_i\}, \tag{20}$$

$$S_i^* = \mathbb{E}\{|R_i - \mathbb{E}\{R_i\}|\}, \tag{21}$$

which is to say that, at the stationary point, $Q_i$ and $S_i$ are the mean reward and mean deviation of reward for arm $i$, respectively. Without using the idealisation, the stationary point is different, but with appropriate parameters still a good representation of mean and spread of the

reward distribution albeit with some additional scaling and bias [13]. Another variation of the learning rule that achieves the exact stationary point given in Eqs 20 and 21 has also been proposed [23], but for the purpose of this study, we are satisfied with using the idealised learning rule given in Eqs 15 to 17. The $Q_i[t]$ variable, like that in the normative strategies, is a dynamically updated estimation of the mean reward. The exact dynamics of this variable in the two implementations is different, since the normative strategies update using Kalman filtering, and the basal ganglia model uses the learning rule derived from the dynamics of direct and indirect pathways. The $S_i[t]$ variable fundamentally differs from $\sigma_i[t]$ in the normative strategies, as it is only an estimation of the spread of reward distribution, whereas $\sigma_i[t]$ is the posterior standard deviation of the mean estimation from Kalman filtering which eventually diminishes with repeated observations. We show later how the basal ganglia model might produce an equivalent $\sigma_i[t]$ variable and use it to inform action selection.

**Effect of dopamine.** Dopamine was found to have opposite modulating effects on the excitability of D1 and D2 neurons [30], increasing that of D1 neurons and reducing that of D2 neurons (Fig 2). Denoting the dopamine level in the striatum as $D_i[t]$, we can thus express the thalamus activity as a result of the activities of the two pathways using

$$T_i[t] = \left(\frac{1 + \lambda D_i[t]}{2}\right) G_i[t] - \left(\frac{1 - \lambda D_i[t]}{2}\right) N_i[t] + eZ, \tag{22}$$

where $\lambda$ is a scaling factor that reflects the strength of dopaminergic modulation. The model assumes here that dopamine level in the circuit has the same modulating effect on the two pathways. $T_i[t]$ is used as the value utility for action selection, much like $V_{\text{UCB},i}[t]$ and $V_{\text{Thompson},i}[t]$ in the normative strategies. Despite this correlation, we will keep using $T_i[t]$ to denote the value utilities derived from the basal ganglia model that can be directly mapped to activity in the thalamus. $G_i[t]$ and $N_i[t]$ in Eq 22 follow the learning rule given above, and $eZ$ is a noise term accounting for all sources of random noise within the circuit. Substituting $G_i[t]$ and $N_i[t]$ with $Q_i[t]$ and $S_i[t]$, Eq 22 is equivalent to

$$T_i[t] = Q_i[t] + \lambda D_i[t] S_i[t] + eZ. \tag{23}$$

From Eq 23, it is easy to arrive at the experimental prediction that actions associated with higher mean reward and higher reward variability would be preferred, and elevated tonic dopamine level in the striatum should lead to higher level of risk seeking in behaviour (i.e. increased tendency of choosing actions with more variable outcomes), and evidence in support of this prediction has been reviewed [13].

## Results

### Dopamine encoding novelty leads to effective exploration

Previously, we introduced a model of the basal ganglia for learning the mean and variability of the reward resulting from an action, which also takes into account the effect of dopamine on the direct and indirect pathways. Next, we make extension to this model to show that a certain dynamics of dopaminergic activity can lead to an implementation of efficient exploration strategy similar to the hybrid strategy introduced before.

Following the reinforcement learning and action selection rules from Eqs 15 to 23, if dopamine level in the basal ganglia circuit stays constant from trial to trial during action selection, actions with higher estimated mean reward (more rewarding on average) and greater reward spread (more risky) are always favoured. This has certain benefits in exploration modulation, especially at the early stages of exposure to a new environment (e.g. at the beginning of a new block of trials in the bandit task). However, as mentioned in Introduction, the posterior

uncertainty of mean reward $\sigma_i[t]$ (rather than reward variability) is the normatively optimal modulator for exploration that can facilitate an effective algorithm throughout the task. Following the central limit theorem and with a neutral prior on the mean reward, this can be represented using

$$\sigma_i[t] = \frac{S_i[t]}{\sqrt{n_i[t]}}, \tag{24}$$

where $S_i[t]$ represents the reward variability and can be updated according to Eq 17, and $n_i[t]$ is the number of times arm $i$ has been chosen up until trial $t$.

With $S_i[t]$ being the sampling standard deviation and each reward drawn from independent and identically distributed random variables, $\sigma_i[t]$ is an representation estimation of the posterior standard deviation, but since both $Q_i[t]$ and $S_i[t]$ are dynamically updated, and at the stationary point $S_i[t]$ gives the absolute mean deviation rather than standard deviation, $\sigma_i[t]$ is not always an unbiased approximation of the posterior standard deviation under this learning rule (see [28] for a different formalisation of the learning rule which mitigates the stationary point issue). Another caveat of this representation of posterior uncertainty is that it is only valid for stationary environments where the reward distributions do not change over time, which the scope of this study is confined to. We come back to this in Discussion.

It is evident now that, in order for the basal ganglia circuit modelled to compute posterior uncertainty, a signal correlated to the sample size $n_i[t]$ is necessary. This is where we formally look at the trial-by-trial variations of dopamine level. While more commonly associated with reward prediction error, transient dopamine activities have also been found to be correlated to novelty in certain reinforcement tasks [15, 16]. Since novelty naturally has negative correlation with the sample size, we make the assumption about the specific form of dopamine level with

$$D_i[t] = (v + \eta Z)\frac{1}{\sqrt{n_i[t]}} \tag{25}$$

or equivalently

$$D_i[t] \sim \mathcal{N}\left(\frac{v}{\sqrt{n_i[t]}}, \frac{\eta^2}{n_i[t]}\right), \tag{26}$$

which is a noisy representation with both mean level and variability negatively correlated to the sample size. $v$ and $\eta$ are constant parameters which scale the effect of novelty on the average and variability of dopamine response, respectively. Substituting Eq 25 into Eq 23 gives

$$T_i[t] = Q_i[t] + \lambda(v + \eta Z)\sigma_i[t] + eZ. \tag{27}$$

This equation shows that when the dopamine is appropriately modulated by novelty, the thalamic activity represents a effective value utility ($V_{\text{hybrid},i}$ in Fig 1D). In the two-armed case, it leads to the choice probability

$$p(c[t] = 1) = p(T_1[t] > T_2[t]) \tag{28}$$

$$= \Phi\left(\frac{Q_1[t] - Q_2[t] + \lambda v(\sigma_1[t] - \sigma_2[t])}{\sqrt{\lambda^2 \eta^2(\sigma_1^2[t] + \sigma_2^2[t]) + 2e^2}}\right). \tag{29}$$

The exploration strategy this basal ganglia model produces shares the same essential property of the hybrid strategy given earlier by Eq 7, in that both the relative uncertainty and total

uncertainty levels affect the choice probability. This model also has isolated UCB and Thompson sampling strategies nested in, which can be recovered when either $\eta$ or $v$ is zero. With all three parameters $\lambda$, $v$ and $\eta$ being independent one is redundant. We address this later with additional constraint based on experimental results.

We have thus shown that an extension of an existing biological model of the basal ganglia yields an exploration strategy with important similarities to efficient normative strategies, that qualitatively matches past experiments. In the rest of Results, we demonstrate the merits of the extended model of the basal ganglia from three perspectives. First, we verify the assumption made in extending the model about the specific mathematical form of the response of dopamine level to novelty using electrophysiological recording data. Next, we compare the fitting to human behavioural data of the exploration strategies from the model with that of the normative strategies. Following this, we show the performance of the basal ganglia strategy in more difficult tasks in comparison with classic UCB strategies.

## Modelling of dopaminergic novelty response

We next present experimental evidence for novelty encoding of dopaminergic neurons, and compare the experimental results with the ideal dopaminergic activity to modulate exploration given by Eq 25.

In an earlier study [14], the response of dopaminergic neurons to conditioned stimuli during the Pavlovian learning task was recorded using electrophysiological recording in awake behaving monkeys. During the experiment, novel reward-predicting visual stimuli, which the animals had never seen before, were presented to animals. Different stimuli were associated with one of three (25%, 50% or 75%) probabilities of reward (a drop of juice). Neural data were collected during the learning task using extracellular single-cell recording of 58 neurons in the ventral tegmental area (VTA) identified as dopaminergic neurons using established criteria. These neurons likely projected to the ventral striatum where D1 and D2 neurons are found [31, 32]. It was shown that the response of the dopaminergic neurons was divided into two distinct temporal phases. The firing rates during the late phase 0.2 to 0.6 s after cue onset differentiated reward probabilities predicted by different stimuli in learned animals. This response pattern is consistent with the theory that dopamine signals reward prediction error. The firing rates during the early phase 0.1 to 0.2 s after cue onset were independent of reward probabilities associated with the cue throughout the experiment even after learning was completed, but decreased as the stimuli causing the response were repeatedly presented, thus reflected stimulus novelty [14]. We focus on the early phase novelty signal here to investigate whether its quantitative form resembles the normatively ideal form given earlier in Eqs 25 and 26.

We performed function fitting on the trial-by-trial evolution of normalised and baseline-subtracted firing rates of dopaminergic neurons during the novelty response phase (Fig 3). The fitting was done using both the average activity of the 58 recorded neurons (Fig 3A), and using individual neuron data with a hierarchical model (Fig 3B and 3C). The functions and fitting methods used are described in more detail in Methods. Numerical values of parameters of functions fitted to average activity as well as fixed and random effects from hierarchical model fitting are given in S1 Table. Best-fitting parameters for each individual neuron from hierarchical model fitting can be found within the supplied code repository. Results from both hierarchical model fitting and fitting to average activity suggest that the inverse square root function (the closest to the normatively ideal form) fits better than the exponential function, but the best fitting function is the power function with three function parameters (Fig 3E and 3F). For

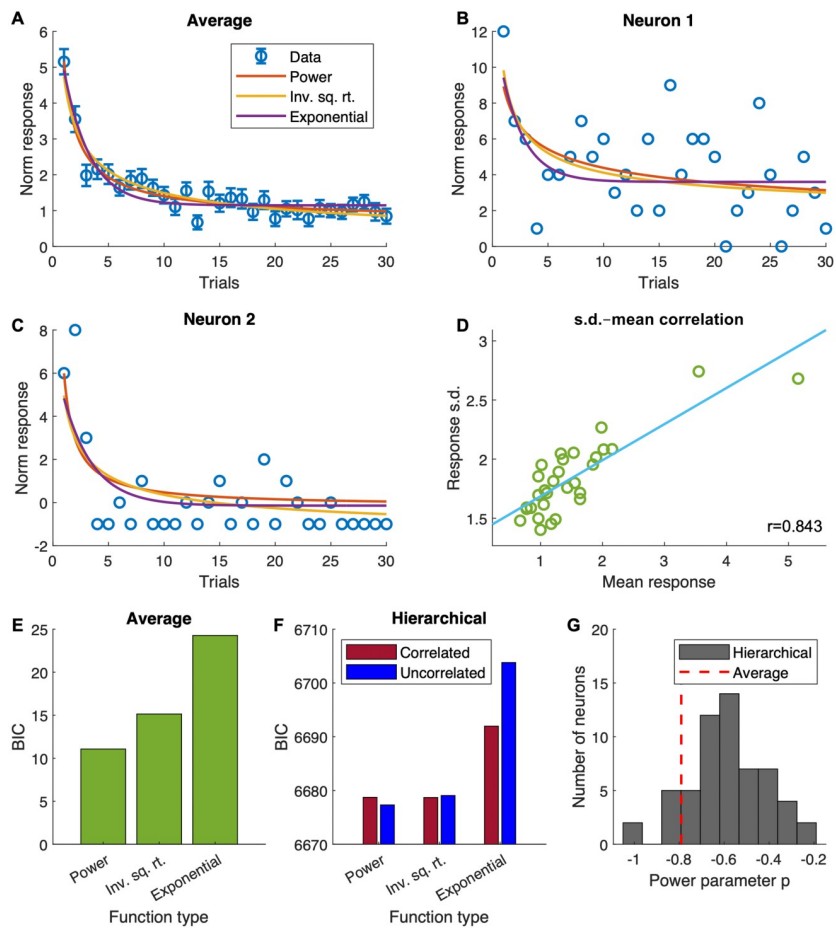

**Fig 3. Results from function fitting to recording data collected during the early phase of the response of VTA dopamine neurons to stimuli during Pavlovian learning. A** Experimental data points of average activity and standard error overlaid with best fitting curves of three different function forms (power function, inverse square root function i.e. power function with power parameter fixed at −0.5, exponential function). **B** and **C** Experimental data points of the activity of two example neurons overlaid with best fitting curves of the same three function forms obtained through hierarchical model fitting. **D** Scatter plot of standard deviation of activity against average activity at each trial overlaid with best fitting straight line (correlation $r = 0.843$). **E** Bayesian information criterion (BIC) values for the fitting of functions to average activity data, showing that the power function is the best fitting. **F** BIC values for fitting hierarchical models (with correlated and uncorrelated model parameters) to individual neurons' recording data, again showing the power function is the best fitting. **G** Power parameter $p$ obtained through fitting to average activity and hierarchical model fitting (shown as a histogram).

fitting using the average activity, the best fitting function is therefore of the form

$$\bar{D}_i[t] = \mathbb{E}\{D_i[t]\} = m + kn_i^\pi[t], \tag{30}$$

and the best fitting power parameter $\pi$ is −0.791 to three s.f., which differs significantly from −0.5 which gives the inverse square root function ($p < 0.05$, two-tailed $t$-test). Also differing from the ideal form is the non-zero intercept $m$ ($p < 0.05$, two-tailed $t$-test). In further analysis, we focus on the fitting using average activity. This allows us to analyse the relationship between novelty and variability of the neuronal responses—the normative analysis (Eqs 24 to 29) show that there should also be positive correlation between novelty and neuronal response variability. Specifically, we assume that the relationship between standard deviation of activities and the number of observations takes identical form as the mean activity. We therefore performed

linear regression analysis on the mean and standard deviation of activities from the 58 recorded neurons (Fig 3D), and found a strong relationship ($r = 0.843$ to three s.f., $p < 10^{-8}$, two-tailed $t$-test). This relationship can be expressed as

$$\text{s.d.}(D_i[t]) = \sqrt{\mathbb{E}\{(D_i[t] - \mathbb{E}\{D_i[t]\})^2\}} = a + b\bar{D}_i[t]. \tag{31}$$

One difference we found between the best fitting function to experimental data and the normatively ideal form is again the non-zero ($p < 10^{-18}$) intercept $a$ in Eq 31.

The electrophysiological recording data used in this analysis was collected during a Pavlovian learning task, and an experiment with a two-alternative choice task was also performed in the same study [14]. We performed a cursory analysis (S1 Appendix) on the partial dataset obtained from this experiment, and reached similar qualitative conclusions.

## Model refinement based on dopamine data

We can now make some amendments to the extensions on the basal ganglia model, utilising the form of dopaminergic novelty response derived from electrophysiological recording data given in Eqs 30 and 31. The modified expression for dopamine level is

$$D_i[t] = \bar{D}_i[t] + (a + b\bar{D}_i[t])Z \tag{32}$$

$$= m + kn_i^\pi[t] + (a + bm + bkn_i^\pi[t])Z, \tag{33}$$

or equivalently

$$D_i[t] \sim \mathcal{N}(m + kn_i^\pi[t], (a + b(m + kn_i^\pi[t]))^2). \tag{34}$$

Mathematically, this is a generalisation of Eqs 25 and 26, with the inverse square root replaced by power function and constant terms added. This leads to an alternative form of posterior uncertainty level (analogous to Eq 24)

$$\hat{\sigma}_i[t] = S_i[t]n_i^\pi[t], \tag{35}$$

which then leads to the output to the thalamus (analogous to the ideal version given in Eq 27) to take the form

$$T_i[t] = Q_i[t] + \lambda((m + (a + bm)Z)S_i[t] + (k + bkZ)\hat{\sigma}_i[t]) + eZ, \tag{36}$$

which gives the choice probability in the two-armed case

$$p(c[t] = 1) = p(T_1[t] > T_2[t]) \tag{37}$$

$$= \Phi\left(\frac{Q_1[t] - Q_2[t] + \lambda m(S_1[t] - S_2[t]) + \lambda k(\sigma_1[t] - \sigma_2[t])}{\sqrt{\lambda^2(a + bm)^2(S_1^2[t] + S_2^2[t]) + \lambda^2 b^2 k^2(\sigma_1^2[t] + \sigma_2^2[t]) + 2e^2}}\right). \tag{38}$$

Compared with the ideal form, there remains a term with $S_i[t]$ which is the result of the additional constant parameters $m$ and $a$. This implies that, according to the choice strategy of the refined model, there is always preference towards actions with more variable outcomes even when there is no uncertainty about the mean rewards, whereas according to Eq 29, the ideal strategy converges towards a (noisy) greedy strategy as uncertainty diminishes. The two strategies are qualitatively similar otherwise. Parameters $a$, $b$, $m$ and $k$ are set to values obtained

from function fitting to neural data, so that there is no redundant parameter in the model fitted to behavioural data.

## Model fitting to behavioural experiment data

We have previously drawn comparison in multiple occasions between the exploration strategies derived from the basal ganglia model and the normative strategies from [12]. While these share common characteristics in their algorithms, we also highlighted some important differences, most notably in the learning rules and the resulting representation of posterior uncertainty. A two-armed bandit task was designed and used in behavioural experiment involving human participants, and the normative strategies were fitted to the behaviour of the participants [12]. It was discovered that the hybrid strategy fitted the data better than isolated directed or random exploration strategies. In this study, we fitted strategies from the basal ganglia model (with parameters describing dopamine novelty response fixed to values estimated above from the activity of dopaminergic neurons) to the same data for an algorithmic level comparison of the strategies. For completeness, we fitted not only the general hybrid strategy defined by Eq 36, but also the special directed and random exploration only strategies.

We used data from one of two experimental setups in [12] (Experiment 2), where both options of the two-armed bandit were associated with stochastic rewards. During the experiment, participants faced twenty blocks of ten trials. Within each block, the rewards from two arms were drawn from fixed Gaussian distributions with different means but identical variances. The mean rewards themselves were drawn from a zero mean Gaussian distribution ($\mathcal{N}(0, 100)$) for each block, and the reward variance of each arm was fixed at 10. The participants were instructed to maximise the total reward over each block. It is worth highlighting at this point that all of the strategies fitted to this dataset are based on the fundamental assumption that the exploration strategy used by the agent remains stationary on a trial-by-trial basis, i.e. the strategy is indifferent to the number of trials remaining. This assumption is mostly valid for this experimental setup.

Trial-by-trial fitting with stochastic maximum likelihood methods was used to obtain optimal parameters of the basal ganglia strategies for each individual participant. Once the optimal parameters were obtained, the corresponding maximum likelihoods were further converted into Bayesian information criterion (BIC) and Akaike information criterion (AIC) statistics used for comparison. These offset the potential benefits brought by extra parameters with different levels of penalty. The strategies fitted to behaviour and methods for fitting are described in more detail in Methods. The best-fitting model parameters for each participant can be found in the supplied code repository, and the mean and variance values of these across participants can be found in S2 Table.

Fig 4 shows comparison of BIC and AIC values from fitting two sets of strategies with different learning rules—one with Kalman filtering as the learning rule from [12], and the other derived from the novel basal ganglia model, with fixed reinforcement learning rates defined in Eqs 18 and 19. Each set consists of four variations with different types of uncertainty-driven exploration (or lack thereof)—the hybrid exploration strategy, the directed and random exploration only strategies, and a "value-only" strategy that does not use any modulated exploration (equivalent to standard Rescorla–Wagner learning for the basal ganglia strategies). As shown in Fig 4A, there is no significant difference in BIC between models with same exploration strategies but different learning rules (except for the value-only models, $p < 0.01$, two-tailed $t$-test). Among models with the same learning rule, the model with hybrid exploration strategy is significantly better fitting than others ($p < 0.01$, two-tailed $t$-test). This confirms the key finding of [12] that humans use a hybrid strategy of directed and random exploration in bandit tasks

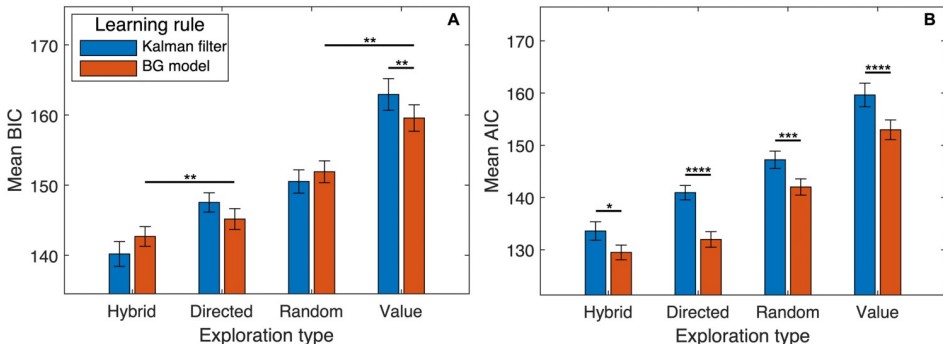

**Fig 4. Model fitting comparison of different behavioural models fitted to human behaviour in Experiment 2 from [12] (a two-armed bandit task).** Error bars show the standard errors computed using normalised data [33], reflecting the within-subject design. The reinforcement learning models fitted employ two different learning rules (Kalman filtering as in [12] and basal ganglia derived learning rule (see Methods for details)). Four models with each learning rule were fitted, each with a different exploration strategy. **A** Mean Bayesian information criterion (BIC) values across participants from trial-by-trial model fitting. Following each of the two learning rules, the model with hybrid exploration strategy gives the lowest BIC value indicating best model fitting, but there is no consistent effect of the learning rule within models of each exploration type according to this metric. **B** Akaike information criterion (AIC) values calculated from the same fitting results. With the reduced penalty for number of parameters in the AIC, the learning rule from the basal ganglia model shows significant advantage.

using a more mechanistic modelling framework based on physiology. This also shows that the exploration strategies derived from the basal ganglia model are similar to the normative strategies with Kalman filtering in terms of their abilities to interpret behaviour at the algorithmic level when a model comparison metric with high penalty for number of parameters such as BIC is used. Given the more idealised learning rule used in the normative strategies that does not account for potential individual differences across participants, one would perhaps expect significantly better fitting from the basal ganglia strategies. However, the effect of individual differences in this task could be relatively small due to the short trial blocks, so that the decrease in BIC from better fitting is outweighed by the increase from additional penalty for extra parameters. This is supported by the AIC values (which include a smaller penalty for number of parameters in this case) shown in Fig 4A, which suggests significantly better fitting of all basal ganglia models compared with their Kalman filter counterparts ($p < 0.05$, $p < 0.0001$, $p < 0.001$, $p < 0.0001$ respectively for the four exploration types in order, two-tailed $t$-tests).

## Performance in simulation of bandit tasks

Variations of UCB strategies have been extensively investigated in analytical studies to assess their performances in multi-armed bandit tasks [6, 10, 11]. As these studies were among inspirations for the current work, we carried out simulations of bandit tasks to compare the in silico performance of the UCB strategy derived from the basal ganglia model with several other types of UCB strategies. These include the variants from [6] as well as the UCB strategy from [12]. We also included the recently proposed OpAL* model [21] in the comparison, since this shares some common features with the basal ganglia strategy in this study. We used the UCB version of the basal ganglia model (without the random exploration element) since the strategies from [6] all have deterministic choice policies, and we wanted to make sure all other strategies in the comparison share this important feature for fairness—this is impossible to achieve in a strategy with random exploration. More details about the models and optimisation protocols can be found in Methods. The optimised parameter(s) for each task and strategy can be found tabulated in S3 Table.

We used five different bandit tasks for the simulations. The first three are the most difficult with $K = 10$ total arms out of which one is marginally more rewarding than the other nine, similar to those used in [6]. These include tasks with either Bernoulli or Gaussian reward distributions (but not both within the same task). The other two tasks were taken from [12] and [21], respectively. The task from [12] involves only $K = 2$ arms with Gaussian reward distributions, and the mean rewards are randomly sampled for each block. The task from [21] has $K = 6$ arms with Bernoulli reward distributions. We tested all strategies in all different tasks, except for the omission of OpAL* in Gaussian bandit tasks since it is undefined in such cases. We follow the convention used in [6] and define the regret at each trial as the difference between the mean reward of the most rewarding action and the mean reward of the chosen action on that trial. This measurement of performance allows for easier comparison across different tasks than the straightforward obtained reward, as it represents the lost reward potential in each task. Importantly, for a given choice, regret is a deterministic quantity while reward may be stochastic, hence regret is a more consistent and less noisy variable in simulations than the reward. By definition, it should always take values between 0 and the difference between the highest and lowest mean rewards of all actions.

Fig 5 shows the results from simulations of the five aforementioned tasks. For each task, regret is plotted over trial number within a block for each strategy. In the relatively easier task from [12] (Fig 5D), all the strategies produced comparable performances with the exception of `UCB1-normal`, which can fail completely in a task with small number of trials such as this

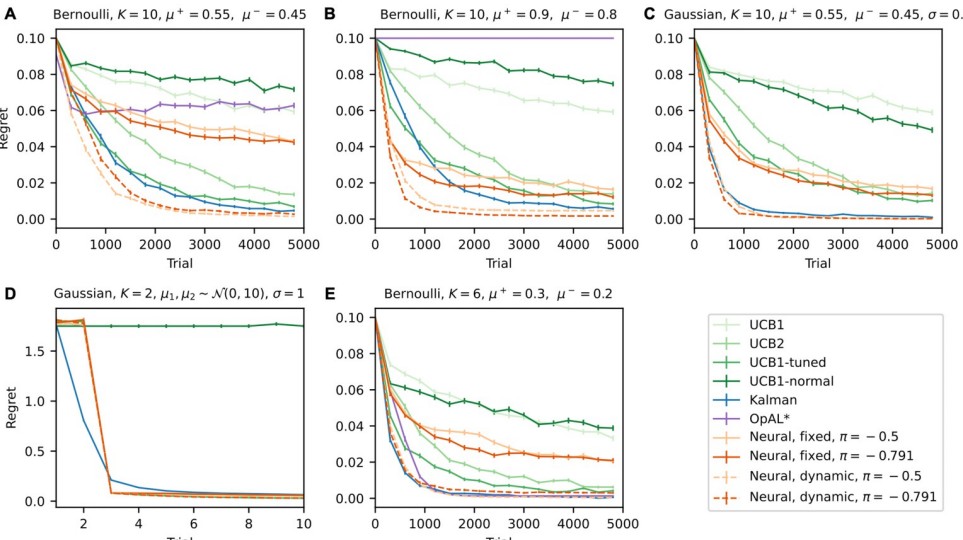

**Fig 5. Performance comparison of neural UCB strategies based on the basal ganglia model against other UCB algorithms in different bandit tasks.** Per-trial regret (defined as the difference between the expected reward of the optimal action and the expected reward of the chosen action at each trial) is plotted against trial number in each panel. Error bars show standard errors over repeated simulations ($N = 20000$ repeated runs for the task in **D**; $N = 1000$ repeated runs for all other tasks). All of the tasks simulated are stationary bandit tasks, such that the reward distribution for each arm remains constant throughout one simulation run. Panel titles describe the individual tasks. Each task has a total of $K$ arms. In all tasks apart from the one in **D**, the reward distributions of all arms remain constant throughout the experiment (across all of the repeated runs). One arm has mean reward $\mu^+$ and all others have mean reward $\mu^-$. For the Gaussian task in **C**, all arms have the same standard deviation $\sigma = 0.3$. The task in **D** has the mean rewards of both arms randomly sampled from the same distribution on each run, and both distributions always have constant variance. We tested neural UCB strategies with two different values of power parameter $\pi$ which are the ideal value predicted by the model ($-0.5$) and the value that best describes neural recording data ($-0.791$). Both fixed learning rate (solid lines) and dynamic (decaying) learning rate (dashed lines) versions of the neural UCB strategies were tested.

one. In the other four tasks, we were able to reproduce the qualitative findings from [6] regarding the classic UCB strategies: the more complex `UCB2` and `UCB1-tuned` perform better than `UCB1` in all experiments; `UCB1-normal` (which is a variation of `UCB1` optimised for Gaussian bandits) performs better than standard `UCB1` only in the Gaussian task. In fact, when comparing directly the tasks in Fig 5A and 5C (which have the same mean rewards for all arms), despite the Gaussian task being less demanding than the Bernoulli task due to smaller reward variances, all strategies from [6] except `UCB1-normal` perform no better in the Gaussian task.

The Kalman filter strategy from [12] consistently outperforms all the classic strategies. It is also able to take advantage of the smaller reward variances in the Gaussian task, and therefore has the most significant advantage against the other strategies in this task. The neural strategies with fixed learning rates (Eqs 18 and 19) have worse performance than the Kalman filter strategy and some classic UCB variants from [6]. Since the most significant difference between the neural strategies and the Kalman filter strategy is in the learning rule, and the Kalman filter is a learning rule with adaptive learning rate, we experimented with variations of the neural strategies with simplified dynamic learning rates defined by

$$\alpha_{i,q}[t] = \alpha_{0,q}\frac{m + kn_i^\pi[t]}{m + k}, \tag{39}$$

$$\alpha_{i,s}[t] = \alpha_{0,s}\frac{m + kn_i^\pi[t]}{m + k}, \tag{40}$$

which gradually reduce the rate of updating mean reward and reward variability estimations as learning progresses, and result in significantly improved performance over the fixed learning rate strategies in all tasks. The improved performance of the neural strategies overall exceeds that of the Kalman filter strategy. Note that in fitting the different models to human behaviour, we did not observe a significant difference between the fixed learning rate neural model and the Kalman filter model, and correspondingly the in silico performance difference in the same task between these exploration strategies is not pronounced (Fig 5D).

We also discovered in analysing neural recording data that the representation of novelty by dopaminergic neurons does not necessarily follow the ideal form the normative model predicts. Here we see that the difference in the specific representation of novelty (i.e. the difference in the value of the power parameter $\pi$) has little effect on the performance of the resulting exploration strategies in simulations.

As mentioned, we also tested the OpAL* strategy from [21], which uses a learning rule that produces estimations of reward distributions to the second moment (similar to the learning rule of the basal ganglia model). This learning rule also includes dynamically adjusted learning rates. The performance of OpAL* according to the regret metric we use varies significantly across the three Bernoulli bandit tasks we tested it in. In the task with $K = 6$ arms and low rewards taken directly from [21] (Fig 5E), OpAL* is among the best-performing strategies; in the two ten-armed tasks (Fig 5A and 5B), we see different levels of degradation in performance, worse in the more rewarding task. In fact, in the task of Fig 5B, this strategy completely fails to pick out the optimal action. As discussed in [21], OpAL* is more exploratory in overall less rewarding environments and more exploitative in more rewarding environments, which is consistent with our simulation results.

Overall, the results of the simulations suggest that a strategy based on the basal ganglia model has overall better performance than the classic UCB strategies and the Kalman filter UCB strategy as well as the OpAL* strategy proposed in [21] in a range of bandit tasks, given

that the learning rate is dynamically adjusted and decays with novelty. However, fixed learning rate strategies do not perform nearly as well.

## Discussion

Our results suggest that the fast transient variations of dopaminergic neuron activity can encode novelty in a way that could contribute to representation of posterior uncertainty in the basal ganglia during reinforcement learning. The uncertainty representation could then be used to facilitate exploration strategies that perform well in simulation and are similar to a normatively ideal construction. We now further discuss the implications of the results and new experimental predictions that can be derived from the model, as well as potential future directions.

### Functions of dopamine in reinforcement learning

The quantitative analysis on the novelty response of dopaminergic neurons made possible by high resolution recording is fundamental to all results from this study. The role of dopamine has always been central in efforts of understanding reinforcement learning. In particular, the transient activity of dopaminergic neurons is widely considered to encode reward prediction errors [15, 34] used to update the predictions of action outcomes. This theory is supported by a plethora of experimental evidence. In fact, the experimental results [14] we analysed also provide support for this theory. The activity of VTA dopaminergic neurons recorded from 0.2 to 0.6 s after cue onset as well as their responses to rewards are highly consistent with the pattern predicted by the reward prediction error theory [14]. Saliently for this work, there has also been observations of correlation between activity of dopaminergic neurons and novelty [15, 16]. This additional variability is often treated as being multiplexed into the reward prediction error signals as a bonus component. Experimental results from [14] provide an alternative view on the multiple factors correlated with transient activity of dopaminergic neurons by observing the different response patterns during the temporal window 0.1 to 0.2 s after cue versus the later 0.2 to 0.6 s window. This suggests the possibility that the novelty and reward prediction error signals are carried by the same dopaminergic neurons yet can still be fully decoupled. Based on this hypothesis, we constructed our reinforcement learning model of the basal ganglia that uses the reward prediction error in belief updates and uses the novelty signal combined with other learned latent variables to modulate exploration in decision making, which is fundamentally different from the "novelty as a bonus" view in many previous models [16]. The dual functionality of fast transient dopamine variations is also supported by evidence uncovered more recently that dopamine conveys motivational value on short timescales and that there exist possible mechanisms for the same target neurons of dopamine to switch between different interpretation modes [35]. The precise mechanism of the two separate functions is beyond the scope of this study, but it could be related to the different lengths of the temporal phases of novelty and reward-encoding activity having qualitatively different effects on striatal neurons.

We found from simulations of challenging multi-armed bandit tasks that learning rates dynamically adjusted according to novelty level can have significant performance benefits. Naturally, this leads to the speculation that the novelty signal delivered by dopaminergic neurons can also modulate the plasticity of corticostriatal connections. Some recent experiments suggest that this mechanism could in fact exist in the brain [36, 37]. From the data used in this study, one could theoretically find conjectural evidence for or against the hypothesis, e.g. by investigating whether the outcome of an "explore" trial (a trial on which the option with lower $Q$ value is chosen, likely to be associated with higher novelty signal) is statistically more

influential on the outcome of the next trial (suggesting a higher learning rate). Within the model framework of reinforcement learning through direct and indirect striatal pathways, a different take on modulated belief updating has been proposed [23], which considers the circuit dynamics at the time of reward presentation and predicts that the reward prediction error itself should be scaled by the estimated spread of the reward distribution (Eq 12). Theoretically, this could be combined with the learning rate modulation by novelty, and from a physiological perspective, the novelty signal should take effect on the target striatal neurons before reward presentation, whereas the dynamics that leads to the scaled prediction error signal occurs after reward presentation.

While the analysis in this work is centred around the transient changes in dopamine level, the tonic dopamine level in the striatum could also influence the circuit dynamics and consequently reinforcement learning behaviour. According to our model, the most significant effect of higher tonic dopamine level should be an overall higher level of risk seeking, and consequently a stronger effect of relative uncertainty on directed exploration. Experimental evidence in support of this prediction was reviewed in [13], and in [17] it was demonstrated that elevated tonic dopamine level resulted in increased novelty seeking (preference towards the less-chosen actions in the context of multi-armed bandit), which can be interpreted as a form of uncertainty preference. However, a more up to date literature contains interesting experimental results that are not necessarily consistent with this prediction. For example, in [38] the correlation between behavioural traits in exploration tasks and single nucleotide polymorphisms of dopaminergic genes was studied, and results show that variation in a gene linked to striatal dopamine predicts the effect strength of relative uncertainty differently than our current model does, in that the genotype associated with higher level of striatal dopamine transmission correlates with reduced effect of relative uncertainty on directed exploration.

There are also several studies suggesting that high level of tonic dopamine reduces random exploration. Genetic disruption of glutamate receptors in dopaminergic and D1 neurons (which reduced dopamine transmission) was found to lead to overall more stochastic and less reward-driven choices [39], while similar effects of reduced D2 receptor occupancy (which also indicated reduced dopamine transmission) have also been found [40]. Further evidence was found through the use of dopamine receptor antagonist [41]. However, it is worth emphasising that our model makes prediction on the effects of dopamine on directed exploration, rather than random exploration, and the opposite effects of tonic dopamine level on these two types of exploration may suggest they rely on fundamentally different mechanisms.

This literature of experimental work highlights the overall complex nature of the influence of tonic dopamine level on reinforcement learning. In this work, we used normalised firing rate data which do not contain any information about the tonic baseline. Correspondingly, our model of the basal ganglia does not explicitly account for the effect of tonic dopamine levels, but the different resulting model parameters across individual participants from fitting model to behavioural data could potentially be correlated to this.

We used electrophysiological recording data obtained during a Pavlovian learning task to study the novelty response of dopaminergic neurons [14]. In this context, novelty is naturally associated with each cue presented in the experiment since there was no action required, whereas the model we are proposing handles reinforcement learning tasks with action selections, and includes a novelty value assigned to each action. In fact, within the same study [14], dopaminergic neurons were also recorded during a two-armed bandit task in which one familiar cue and one novel cue (and actions associated with each) were present. The recorded neurons had novelty response similar to that observed in the Pavlovian learning when the novel action was taken (see S1 Appendix for analysis). During this task, animals typically made saccade actions between 0.6 to 0.65 s after cue onset, which was better aligned with the reward

predicting phase of dopaminergic activity rather than novelty encoding phase. According to our model, the novelty signal is involved in action selection, while the subsequent reward-correlated signal is used for learning. It is possible that the delay between novelty signal and action is the time required to integrate the signals for different actions and form decision and action; during this time, an internal prediction of the action outcome is also formed and encoded by the reward-predicting activity.

An important assumption made by our model is that the novelty level of each individual available action can be independently represented. We unfortunately do not have sufficient data to verify this in the current study. Based on previous investigation on the morphology of axonal projection of dopaminergic neurons [42], each dopaminergic neuron can project to thousands of neurons in the striatum, whereas the number of dopaminergic neurons that can influence each striatal neuron is lower by orders of magnitude. This shows that the level specificity of dopaminergic projection is not particularly strong, but should still theoretically be sufficient to support the mechanism proposed by our model.

## Alternative theories of exploration modulation in the brain

The model we propose in this work suggests that the basal ganglia are responsible for both learning the associations of high-level actions with resulting rewards and using this information to select actions following near-optimal strategies. A related model from [20] also highlights the role of the basal ganglia in decision making while describing the relevant circuit dynamics in more detail. These authors made experimental predictions about the effect of tonic dopamine level on the level of random exploration, which suggest that an increase in dopamine level should generally lead to more exploitative behaviour. This qualitatively differs from what our model would suggest, and is supported by some but not all related experimental evidence as discussed previously. Another model of exploration modulation in the basal ganglia (OpAL*) was proposed in [21] which includes a description of the trial-by-trial variation of dopamine level at action selection. Instead of a simple novelty signal, these authors proposed a "meta-critic" mechanism that learns the overall reward level of the entire environment and controls the dopamine level at action selection accordingly. This results in more exploratory behaviour in overall "richer" environments. This meta-critic also operates on a longer time-scale compared to the type of dopaminergic dynamics in the model we propose in this study. The authors also compared in silico performance comparison which put their model ahead of classic UCB strategies. However, this model is only defined for Bernoulli bandit tasks that produced binary reward outcomes in experiments and simulations. The mean reward and reward standard deviation following a Bernoulli reward distribution are always correlated, and it is not trivial what effect this feature had on the conclusion reached by the authors. A continuous reward distribution is a more realistic representation of real-world scenarios, and we have shown in this work that the exploration strategy based on our basal ganglia model can effectively modulate exploration and exploitation in a bandit task with continuous (Gaussian) reward distribution. We also showed in our own simulations that OpAL* does not have the best generality even in Bernoulli bandit tasks compared to some of the other algorithms we tested.

Our analysis of the novelty response of dopaminergic neurons suggests that the way novelty is encoded in dopamine level could be the source of a hybrid strategy of directed and random exploration. Given that this type of strategy is prominent in behaviours, there have also been prior studies looking for the underlying mechanism in the brain. Some of these studies found significant correlations between exploratory behaviour and activity in certain cortical regions, and specifically found cortical regions that are linked with only one of directed or random

exploration but not the other [43, 44]. These theories on the role of the cortex in exploration are mostly beyond the field of view of this study, but it is of course entirely plausible that the basal ganglia are not the sole source of control over exploration modulation.

### Experimental predictions

The model presented in this study allows us to make experimental predictions both on the algorithmic and neural implementation levels. On the algorithmic level, the setup of behavioural task from [12] is not the most suitable for a study comparing the fitting of all behavioural models in this study. Longer trial blocks in more challenging tasks would be more capable of distinguishing between the different learning rules, and having different reward variances both for different options in the same block and from block to block would result in an overall richer dataset and also prevent the subjects forming a prior on the variances over multiple blocks which is a potentially existing phenomenon the current models do not account for. In sum, a task design with both the mean rewards and reward variances for each option randomly chosen for each block of trials would theoretically be the best at revealing the learning dynamics and choice strategy at the algorithmic level.

At the implementation level, the most interesting next step would be to directly verify the role of transient variations in dopamine level in exploration modulation. This would need to involve manipulation of the activity of dopaminergic neurons with high temporal precision relative to option presentation during a multi-armed bandit task. Specifically, manually inducing a short temporal period of high dopamine release in the striatum right after presentation of options (within 0.2 s) should lead to higher tendency of exploration (risk seeking) in the action selection that immediately follows. Since this action selection occurs before any belief update can happen, any such effect can only be the result of exploration modulation but not learning. Conversely, inhibition of dopaminergic signals within the same temporal window should lead to stronger exploitative tendency in the following action selection. The strength of the effect of this manipulation should also vary with the spread of the reward distribution, since this is combined with the novelty signal to produce the posterior uncertainty according to our model.

As previously mentioned, the assumption that the novelty levels of multiple available actions can be independently represented via individual dopaminergic channels remains to be tested. The most promising approach to verify this is to record simultaneously from multiple individual dopaminergic neurons during a choice task, and look for evidence that these neurons can in fact independently track the novelty levels of different actions. This method can be complemented by measurements of dopamine level in the striatum (e.g. through photometry), which could further reveal whether the potentially independent novelty signals remain separable at axon terminals.

For the basal ganglia to facilitate a hybrid exploration strategy, the variability of the transient novelty response of dopaminergic neurons as well as the mean response needs to be modulated (Eq 34). The source of this variability is currently ambiguous. The mechanism most consistent with the model would involve a large number of dopaminergic neurons projecting to each striatal neuron, and only one or a few taking effect on any given trial. This does not seem very realistic, and yet another unlikely requirement of this setup is that somehow the relevant D1 and D2 striatal neurons encoding for the same option need to read out from the same dopaminergic neurons on each trial. A somewhat more likely assumption is that the trial-by-trial variability of the same dopaminergic neurons projecting to each striatal neuron facilitates the sampling. This also does not require the unrealistic assumption that related D1 and D2 neurons always selectively receive from the same dopaminergic neurons, but still

requires the dopaminergic neurons projecting to them to have the same upstream source, which is nevertheless much more reasonable. Given the large differences in functions fitted to individual neurons even when using normalised firing rates (Fig 3B and 3C), it is tricky to build a completely rigorous model based on this assumption since additional scaling would be required, but the key properties of the model should remain the same. Available experimental data from [14] does not particularly support any one of these assumptions over the other, since neurons were recorded one at a time and each neuron was recorded only over one block of trials. Simultaneous recording from multiple dopaminergic neurons that respond to the same cue would be the most effective method. Any correlation between the deviations of activity from their respective fitted functions would be strong evidence for the second assumption above.

## Future directions

The model proposed in this study can be further refined and extended to address two fundamental constraints that currently exists. Firstly, the multi-armed bandit task is a stationary task, which is to say that the reward distributions of all options always remain constant within each block, and the agent always has perfect knowledge of when the contingency changes occur at block crossovers. When the task is generalised to a non-stationary multi-armed bandit, the monotonic novelty representation by dopaminergic neurons is clearly no longer optimal as it can strongly bias the estimation of posterior uncertainty. An abrupt contingency change leads to a fast transient increase in the estimated reward variability according to the learning rules of our model, and from a normative perspective, this is certainly a marker that could be used to trigger a reset or adjustment of the novelty representation and therefore minimise the bias in estimation of posterior uncertainty. On the other hand, the shortcomings of the current learning rule would be harder to remedy when the non-stationarity takes the form of a continuous graduate drift in the reward distribution. The Kalman filter as a learning rule is better suited to tackle these non-stationary tasks. A recent study [45] proposed an exploration model that uses Kalman filter in non-stationary bandit tasks, and showed that the Kalman filter, given parameters suitably adjusted to the task environment, can achieve effective learning capable of optimal modulation of exploration in these tasks. The alternative learning rule also based on the GO-NOGO basal ganglia model proposed in [23] uses scaled reward prediction errors, and can approximate a Kalman filter in certain non-stationary tasks without the need to have parameters tuned specifically to the task environment. A meta-learning mechanism that also addresses the generality issue with the Kalman filter was proposed in [46]. Other models with variable learning rate such as the adaptive learning rate models in [47, 48] also show significant advantage in their adaptability in non-stationary environments. Beside these normative discussions, recency representation in the brain [49] (which can be considered as novelty representation that discounts observations from a long time ago) could potentially be used to further develop the basal ganglia model and improve the performance of its algorithm in non-stationary tasks, and the resulting algorithm might also serve as a solution to the generality issue of Kalman filter. In sum, there are multiple avenues that can be explored in the future which can help with better understanding of basal ganglia functions in non-stationary reinforcement learning tasks.

Secondly, given the first constraint is satisfied, the agent should employ a stationary strategy in respect of the trial number within a block. This could be violated when a situation with a known and very limited number of trials are left before a contingency change, and there are still high uncertainty levels associated with some of the actions. In such scenarios, exploratory behaviour could give way to risk aversion. This is a possible but unlikely occurrence in the

experiments of [12] due to the relatively small reward variability. This phenomenon was investigated in [3], but a mechanistic model is yet to be developed. Since this mechanism would involve dynamics on a longer timescale, we could potentially look for a shift in the tonic dopamine level as a contributor once the model is expanded to account for its effect.

## Summary

In conclusion, the model we propose in this study provides novel insights on how effective exploration strategies could be achieved in the brain, specifically the basal ganglia. Experimental results in support of the model were analysed, and simulations indicate performance merits of the model algorithm. Interesting experimental predictions are derived from the model, and we expect future work to verify the new predictions and to further refine the model for greater levels of detail and better generality.

## Methods

### Function fitting to neural recording data

The neural recording data used for function fitting is in the form of normalised and baseline-subtracted average firing rate over the fixed-length temporal window after cue onset. Normalisation is performed by dividing the raw firing rate during the measurement period by a reference firing rate taken immediately before cue onset.

Three different functions were fitted to the novelty response of dopaminergic neurons. The inverse square root function with two free parameters:

$$f(n) = m + \frac{k}{\sqrt{n}}. \tag{41}$$

The power function with three free parameters:

$$f(n) = m + kn^{\pi}. \tag{42}$$

The exponential function with three free parameters:

$$f(n) = m + ke^{\pi n}. \tag{43}$$

Two different techniques were used for model fitting. First, the average activity of all recorded neurons at each given trial number was computed, and maximum likelihood fitting of three generative models was done on the average activity using MATLAB function `fminsearch`. Bayesian information criterion (BIC) statistics were then computed manually using the resulting maximum likelihood values. Second, hierarchical mixed-effects models were fitted to individual neurons' recording data using MATLAB's `nlmefit` function. BIC values were returned directly by the function. The population distribution of model parameters were modelled both as a fully joint distribution and independent distributions of each of the free parameters. Numerical results of function fitting can be found in S1 Table.

### Model fitting to behavioural data

Eight different reinforcement learning strategies were fitted to behaviour of human participants. These differ in two dimensions: learning rule and exploration type. Two learning rules and four exploration types were tested, giving the total of eight models. One learning rule is derived from the basal ganglia model and the other is the Kalman filtering as described in [12]. A full list of relevant equations that define the strategies and the free parameters that were fitted to behaviour are listed in Table 1. Note that the value utility function for random

exploration (Thompson sampling) strategies with basal ganglia model-derived learning rules is not nested within Eq 36 (since these strategies are not realistic according to the results of our neural data analysis—they are included for completeness only). The value utility for them is given by

$$T_i[t] = Q_i[t] + \lambda(((a + bm)Z)S_i[t] + bkZ\hat{\sigma}_i[t]) \tag{44}$$

with the additional constraint $\lambda > 0$. The Kalman filter-based strategies used as a baseline and the methods used for fitting were described in detail in [12].

Trial-by-trial model fitting of the strategies derived from the basal ganglia model was done using MATLAB's `fmincon` function. Each individual participant were independently fitted with a unique set of optimal parameters. Maximum likelihood fitting was used, with the choice likelihood computed using the value utility function at each trial, and the sum-log-likelihood for each individual participant maximised. The optimiser function was run repeatedly with 50 different initial guesses, and the best results out of the repeated runs were taken. Initialisation of latent parameters followed the same protocols of those used in [12].

## Models used in simulation

We compared the performance of several different directed exploration (UCB) strategies in simulation using a variety of bandit tasks. Specifically, we used a series of computationally efficient strategies detailed in [6] as well as the Kalman filter-based strategy [12] and neural-inspired strategies derived from the basal ganglia model. We also used the OpAL* strategy recently proposed in [21], since it utilises a similar learning rule as our basal ganglia model. The Kalman filter strategy, OpAL* and neural strategies were initialised with mean reward and standard deviation estimators all at 0.5 for all tasks apart from the Gaussian task from Experiment 2 of [12] (Fig 5D), where they were initialised at 0 instead. This differs from the initialisation used in [12] which assumes more knowledge about the task. Otherwise, the Kalman filter strategy follow the same description given previously in Table 1, but with $e = 0$ to make the action selections deterministic (since we are comparing here against deterministic strategies from [6]). The free parameter $\theta$ was optimised for each task. The neural strategies inspired by the basal ganglia model also largely follow the descriptions given in Table 1, except all with fixed parameters $a = b = m = 0$. $k$ then becomes a redundant parameter and is fixed to 1. Noise level $e$ is also set to 0, same as for the Kalman filter strategy. $\pi$ is set to either −0.5 (the value giving optimal reward posterior estimates) or −0.791 (the value obtained from experimental data). The remaining free model parameters $\alpha_q$, $\alpha_s$ and $\lambda$ were optimised for each of the tasks with a crude global minimisation search to minimise the total regret accumulated over all trials. In addition, we also fitted variations of the neural strategies with dynamically adjusted learning rates as described in Eqs 39 and 40, in which cases the initial learning rate parameters $\alpha_{0,q}$ and $\alpha_{0,s}$ were optimised instead of $\alpha_q$ and $\alpha_s$. The OpAL* strategy follows the construction and parameter optimisation protocol described in [21], again with the exception that the parameter controlling choice stochasticity is fixed to give a deterministic action selection policy. This protocol fixes three parameters and performs a grid search over the remaining to minimise total regret accumulated over trials. Values of the optimal parameters can be found in S3 Table.

In our simulations, we did not change the "physical location" (or in coding terms, the location within the array) of the more rewarding arm between repeated runs (this does not apply to the task from [12] in Fig 5D). This implementation choice does not have any major influence on the results, as the simulated agent had no way of learning this higher-level structure; the only artefact resulting from it appears in the task in Fig 5B, where OpAL* failed to pick out

**Table 1. Full description of strategies fitted to behavioural data.**

| Learning rule | Exploration type | Fixed parameters | Fitted parameters | Equations |
|---|---|---|---|---|
| Kalman filter | Hybrid | N/A | $\gamma, \theta$ | 7 |
| | Directed | | $\theta, e$ | 1 |
| | Random | | $\gamma$ | 4 |
| | Value | $\theta = 0$ | $e$ | 1 |
| Basal ganglia | Hybrid | $a = 1.380, b = 0.306,$ $m = 0.677, k = 4.486,$ $\pi = -0.791$ | $\alpha_q, \alpha_s, \lambda, e$ | 15, 16, 17, 18, 19, 36 |
| | Directed | $a = 0, b = 0,$ $m = 0.677, k = 4.486,$ $\pi = -0.791$ | $\alpha_q, \alpha_s, \lambda, e$ | |
| | Value | $a = 0, b = 0,$ $m = 0, k = 0,$ $\pi = 0, \alpha_s = 0$ | $\alpha_q, e$ | |
| | Random | $a = 1.380, b = 0.306,$ $m = 0.677, k = 4.486,$ $\pi = -0.791, e = 0$ | $\alpha_q, \alpha_s, \lambda$ | 15, 16, 17, 18, 19, 44 |

Each row correspond to one of the models. The fixed parameters are determined either by model constraints or analysis of neural recording data. The fitted parameters are the free parameters fitted to the behaviour of individual participants. Equations are the indices of equations in previous text that describe the models. Note that for the Kalman filter models, the equations cited only describe the action selection but not learning through Kalman filtering. For a full description of these models, see [12].

the optimal action in every single run, rather than picking it out by chance at least once every ten runs.

## Supporting information

**S1 Appendix. Cursory analysis of recording data from choice task.**
(PDF)

**S1 Table. Parameters of models fitted to neural recording data from the Pavlovian task from [14].** As described in the main text, we fitted functions to the mean activity of all neurons recorded during the Pavlovian learning task, and also utilised the activity of individual neurons to perform hierarchical model fitting. For fitting using average activity, the best fitting parameters of each function are given as row vectors. For hierarchical model fitting using activity of individual neurons, the fixed effects are given as row vectors and the random effects given as covariance matrices. The equations of the fitted functions are given in Methods.
(PDF)

**S2 Table. Numerical results from model fitting to behavioural data from [12].** Mean and variance values across participants of each fitted parameter are given. The definition of models and values of fixed parameters used are given in Table 1.
(PDF)

**S3 Table. Optimised parameters for exploration strategies used in simulations.** Optimal parameter values (rows) for each of the five tasks (columns) used in simulations. The column titles correspond to the figure panels in Fig 5. Note that OpAL* was only studied in Bernoulli bandit tasks due to its limitation.
(PDF)

## Acknowledgments

The Authors thank Moritz Möller for discussion.

## Author Contributions

**Conceptualization:** Rafal Bogacz.

**Data curation:** Yuhao Wang, Armin Lak.

**Formal analysis:** Yuhao Wang.

**Investigation:** Yuhao Wang.

**Supervision:** Sanjay G. Manohar, Rafal Bogacz.

**Writing – original draft:** Yuhao Wang.

**Writing – review & editing:** Armin Lak, Sanjay G. Manohar, Rafal Bogacz.

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
