## [Decision Letter · Decision Letter 0]

18 Oct 2023

Dear Prof. Bogacz,

Thank you very much for submitting your manuscript "Dopamine encoding of novelty facilitates efficient uncertainty-driven exploration" for consideration at PLOS Computational Biology.

As with all papers reviewed by the journal, your manuscript was reviewed by members of the editorial board and by several independent reviewers. In light of the reviews (below this email), we would like to invite the resubmission of a significantly-revised version that takes into account the reviewers' comments.

We cannot make any decision about publication until we have seen the revised manuscript and your response to the reviewers' comments. Your revised manuscript is also likely to be sent to reviewers for further evaluation.

Sincerely,

Kim T. Blackwell, V.M.D., Ph.D.

Academic Editor

PLOS Computational Biology

Daniele Marinazzo

Section Editor

PLOS Computational Biology

Reviewer's Responses to Questions

**Comments to the Authors:**

Reviewer #1: In this paper, Wang and colleagues show how a well-validated model of dopaminergic effects on the basal ganglia can account for the relationship between dopamine and exploration. The model's empirical predictions were tested on behavioral and neural data, and compared against several benchmark models.

Overall, I think this is an interesting paper, well-written and clearly argued, though the data analyses don't speak decisively to the superiority of their model (see below). I've been hoping for a while that somebody would undertake such a synthesis of Gershman's cognitive models with Bogacz's biological models. I also liked how the authors investigated parametric deviations from their strong modeling assumptions.

Major comments:

- Eq. 24 is derived under the assumption that samples are IID. I don't think this is mentioned anywhere explicitly? This diverges from the more general assumptions of the Kalman filter, as in Gershman (2018) and subsequent related work.

- The idea of dopamine scaling inversely with novelty is very interesting. I think it might be worth pointing out that this is not by itself sufficient to capture all the behaviors of a Bayesian model like the Kalman filter. In particular, Fan et al. (2023, Nature Human Behaviour) studied a version of the two-armed bandit where the underlying means diffused over time at different rates. In this case, the sample size could be the same (depending on how often each arms was chosen) but the posteriors would still be different. That paper showed evidence for sensitivity to this form of "epistemic uncertainty" as predicted by the Kalman filter. I'm curious how the authors think about this finding in the context of their model.

- I think the major limitation of the paper is that it doesn't present direct evidence for the model's neural prediction in the context of an instrumental task that involves exploration. The only dopamine data analyzed here is from a Pavlovian task, where there are no choices.

- My understanding is that lambda is constrained to be positive (though this isn't mentioned explicitly in the paper). This assumption, together with Eq. 29, implies that in the regression analyses reported by Gershman (2018) there should be a negative correlation between the coefficients for the relative and total uncertainty terms. To see this, consider what happens if you simluate data from the model for different levels of lambda, but fit the data assuming lambda=1 (as in Gershman's analysis). Higher values of lambda will inflate the relative uncertainty coefficient and deflate the total uncertainty coefficient. I was curious about this, so I looked at the parameter estimates from Gershman's paper. For experiment 1, the correlation was very close to 0. For experiment 2, the correlation was strongly *positive* (r = 0.68, p < 0.0001). I wonder what the authors think about this. More broadly, I'm very interested in whether the authors can pinpoint aspects of their model that make distinctive behavioral predictions relative to previously reported models.

Minor comments:

- I'm not sure why there's a comma in "V_{UCB,i}[t]" and "V_{Thompson,i}[t]".

- Fig 4 caption: "trail-by-trial" -> "trial-by-trial"

- What do the error bars show in Fig 4? Please note this in the caption. I suspect that these are between-subject error bars, when what you want here is within-subject error bars that reflect the paired comparisons.

- p. 19: "[36] found that stronger striatal dopamine transmission reduced the effect of relative uncertainty on directed exploration." I would word this differently, since that study looked at single nucleotide polymorphisms, not dopamine transmission. There is a connection but it is indirect.

- Gershman (2018) reported two experiments. Which ones did you fit?

Reviewer #2: Based on the models (Mikhael & Bogacz 2015; Moller et al. 2022) that proposed the D1 and D2 MSNs learn positively and negatively biased reward prediction, this article presents how the uncertainty of the mean of the reward can be estimated, by taking into account the number of samples, and how that can be used for directed exploration.

Authors focus on 1) linking the biological model of basal ganglia with the normative cognitive model by assuming the specific form of dopamine level (eq.25); 2) analyzing the electrophysiological data to verify if dopaminergic activities at VTA signals the novelty and follows the hypothetical form; and 3) showing the biological model of basal ganglia explains the behavior better and outperforms variations of UCB strategies when the learning rate is dynamically varying.

The proposed model is interesting and worth further experimental examination. There are some points that need to be clarified before publication.

1) Multiple roles of dopamine: This paper proposes several different effects of dopamine in different timings.

a) plasticity D1 and D2 MSNs by eq. (9) and (10)

b) modulation of thalamic activity by direct/indirect pathways by eq. (22)

While a) is supposed to be for the timing after action selection, b) is for the timing of action selection.

Experimental data of Lak et al. (2016) shows novelty-dependent responses within 0.1 to 0.2 sec. from cue onset. Action selection, however, may not happen within that time frame. Subjects may choose an action with a longer delay, or even spontaneously without any cue.

In the choice task of Lak et al., the saccade reaction time was about 50 ms shorter in early exploratory trials, but it was around 600 to 650 ms. The dopamine response had two peaks; first at 0.1-0.2 sec which was novelty dependent and second at 0.4-0.65 sec, which was value coding. It appears that the second component is more relevant for the action choice. This link between the very early novelty-dependent dopamine and action selection that can happen later should be better explained.

2) Meaning of i: In the mathematical formulation of eq. (22) and onwards, i should be the index of neural population coding a certain action, and the dopamine activity D_i is also supposed to be different for different cues or actions.

An important question is whether dopamine neurons can simultaneously carry different novelty signals for different cues or actions, given the wide-spread nature of dopamine projection, e.g., in Matsuda et al. (JNS, 2009). It might be that dopamine neurons respond to the most novel cue, as in Lak et al., but a question is how that can affect the value of the most novel cue.

3) Performance in simulation: Figure 5 presents the simulation results for a virtual experiment with 10 actions over 5000 trials, but it is more helpful if such a simulation is done for experiments with animal or human data.

Most importantly, for the dataset of Gernman, in addition to model comparison by BIC, it is interesting to compare the simulated performances of different models with the fitted parameters. It is also interesting to do the same for the choice task of Lak et al.

Detailed comments:

In line 53, the description of the multi-armed bandit task can be more concrete. In line 111, the authors describe this task again. It might be better to put two parts together.

In line 62, the authors wrote, “If an agent follows a greedy strategy that does not involve any exploration at all and always prefers the optimal action according to current knowledge, they would simply play each arm exactly once at the beginning of each block of trials.” If the agent also plays greedily without any prior knowledge, shouldn't it stick to the option that gives the first reward?

In section 2.1, it would be clearer to state this study assumes the environment is stationary before further discussion.

For eq. (25), the meaning of eta and nu should be explained.

After model fitting, the authors refine the DA response from eq. (24) to (35) and the value signal from (27) to (36). Is it possible to explicitly describe the choice probability given by this refined model as eq. (29) and discuss whether the refined version shares those essential properties as well.

In section 2.4, the authors explain that “the decrease in BIC from better fitting is outweighed by the increase from additional penalty for extra parameters.” To support this statement, authors should also report the fitting result before penalization by parameters, e.g. average (log) likelihood.

For performance comparison in section 2.5, in addition to the regret, the actual performance and the frequency of exploratory choices may also be better reported.

The authors found that the neural strategies with fixed learning rates were worse than normative strategies and then used the neural strategies with varying learning rates. It is not surprising that a model with time-varying parameters has better performance. To show the efficiency of neural strategies, it is better to introduce other models with time-varying parameters and compare their performance.

In section 3.1, authors mentioned the risk-seeking and novelty-seeking behavior, which appeared several times before. Authors should make a clear difference between risk and novelty in relevant discussion.

Reviewer #3: Yuhao Wang, Armin Lak, Sanjay G. Manohar and Rafal Bogacz present a novel basal ganglia model whereby direct and indirect striatal pathways act together to estimate both the mean and variance of reward distributions, and mesolimbic dopaminergic neurons provide transient novelty signals, facilitating effective uncertainty-driven exploration. The work is composed of model simulations and comparisons with classic exploration algorithms like UCB, model fitting to behavioural data, and model simulation to reproduce electrophysiological data. Their results show that their neurophysiologically detailed and plausible exploration model can approximate the normative exploration strategies while at the same time explaining a set of electrophysiological and behavioural data. Overall, this suggests that transient dopamine levels in the basal ganglia that encode novelty could contribute to an uncertainty representation which efficiently drives exploration in reinforcement learning.

This paper presents an elegant formalization of uncertainty-based, novelty-based and random exploration mechanisms in the basal ganglia. The basal ganglia model is built in a way that encapsulates normative exploration strategies (UCB and Thompson sampling). While further analyses, simulations and clarifications are needed to make the paper stand alone, there is the potential for a nice contribution to the field.

Below are specific comments and questions to help improve the paper.

There is a slight ambiguity in the adopted definition of exploration: "In this study, we generalise from real-world scenarios and define exploration to be any behaviour by a learning agent that favours actions which are sub-optimal in terms of their expected rewards according to the current best knowledge, and exploitation as behaviour that chooses the optimal action with highest expected reward."

It is not clear if the "current best knowledge" is from the point of view of the agent or from the point of view of an ideal observer, which would lead to different labeling of exploratory trials.

Modulated exploration does not necessarily mean directed exploration. See for instance Velentzas et al. 2017 RLDM for a comparison of classical UCB methods with an algorithm instantiating active tuning of random exploration rate. Importantly, such a dynamic random exploration rate enables fast adaptation to non-stationarity, similar to the dynamic learning rate models discussed lines 626-629.

Lines 90-92 "Based on this assumption, tonic dopamine level in the striatum can influence the overall level of risk seeking in behaviour because of the opposite effects dopamine has on D1 and D2 neurons." This is precisely the hypothesis put forward by Humphries et al. (2012) Frontiers in Neuroscience. However, Humphries and colleagues made the assumption that the role of tonic dopamine is opposite to the one presented here that "higher dopamine level should result in a stronger preference for more risky actions with more variable outcomes." In Humphries et al. (2012) such a positive relation between dopamine and exploration could be obtained only with specific parameters determining the strength of basal ganglia control over target structures (blue area in their Figure 6A), which they argued could correspond to specific basal ganglia territories. However, they assumed that the most common effect of tonic dopamine in the basal ganglia on exploration should be negative: more dopamine should produce more exploitation. This topic is partially discussed in the present paper. Nevertheless, some further discussion would be useful. Can the difference be due to different connectivity patterns in different basal ganglia territories? Is it due to different species (rodents, primates)? Is it due to different mechanisms relying on tonic versus transient versus phasic dopamine signals? How many papers found experimental evidence in each different species in favor of positive versus negative influence of dopamine on exploration?

Please cite Collins AGE and Frank MJ (

---

## [Decision Letter · Decision Letter 1]

23 Mar 2024

Dear Prof. Bogacz,

We are pleased to inform you that your manuscript 'Dopamine encoding of novelty facilitates efficient uncertainty-driven exploration' has been provisionally accepted for publication in PLOS Computational Biology.

Before your manuscript can be formally accepted you will need to complete some formatting changes, which you will receive in a follow up email. A member of our team will be in touch with a set of requests.  You also need to make your code publicly available.

Best regards,

Kim T. Blackwell, V.M.D., Ph.D.

Academic Editor

PLOS Computational Biology

Daniele Marinazzo

Section Editor

PLOS Computational Biology

Reviewer's Responses to Questions

**Comments to the Authors:**

Reviewer #1: I'm satisfied by the response to my comments.

Reviewer #2: I am disappointed that this revision added som excuses but did not add any substantially new work. The proposed mechanism of dopamine projections selectively affecting the neurons encoding the action i is impractical. Nevertheless, the idea is interesting and worth sharing in the community as a basis for further developments of biologically realistic models of biological reinforcement learning.

Reviewer #3: The authors have addressed all my concerns.

**Have the authors made all data and (if applicable) computational code underlying the findings in their manuscript fully available?**

Reviewer #1: **No: **The authors say they will share it at the time of publication.

Reviewer #2: Yes

Reviewer #3: **No: **The said they will "at the time of publication".

PLOS authors have the option to publish the peer review history of their article (what does this mean?). If published, this will include your full peer review and any attached files.

Reviewer #1: No

Reviewer #2: No

Reviewer #3: No

---

## [Editor Report · Acceptance letter]

6 Apr 2024

PCOMPBIOL-D-23-01473R1 

Dopamine encoding of novelty facilitates efficient uncertainty-driven exploration

Dear Dr Bogacz,

I am pleased to inform you that your manuscript has been formally accepted for publication in PLOS Computational Biology. Your manuscript is now with our production department and you will be notified of the publication date in due course.

With kind regards,

Zsofia Freund
